# Resistance Training in Breast Cancer Survivors: A Systematic Review of Exercise Programs

**DOI:** 10.3390/ijerph17186511

**Published:** 2020-09-07

**Authors:** Leidy Sofía Montaño-Rojas, Ena Monserrat Romero-Pérez, Carlos Medina-Pérez, María Mercedes Reguera-García, José Antonio de Paz

**Affiliations:** 1Escuela Normal Superior of Pitalito, 417037 Pitalito, Colombia; 2Department of Sports Science and Physical Activity, University of Sonora, 83000 Hermosillo, Mexico; 3Sciences Health School, University Isabel I, 09003 Burgos, Spain; carlosmedinaper85@gmail.com; 4SALBIS Research Group, Faculty of Health Sciences, Universidad de León, 24071 León, Spain; mercedes.reguera@unileon.es; 5Institute of Biomedicine (IBIOMED), University of Leon, 24071 Leon, Spain; japazf@unileon.es

**Keywords:** breast neoplasms, exercise prescriptions, strength, physical function, quality of life, lymphedema

## Abstract

The aim of this study was to identify the characteristics of resistance training (RT) programs for breast cancer survivors (BCS). A systematic review of the literature was performed using PubMed, Medline, Science Direct, the Cochrane Breast Cancer Specialised Register of the Cochrane Library, the Physiotherapy Evidence Database (PEDro), and Scopus, with the aim of identifying all published studies on RT and BCS from 1 January 1990 to 6 December 2019, using the Preferred Reporting Items for Systematic Reviews and Meta-Analyses (PRISMA) guidelines. The risk of bias in the studies was assessed using the revised Cochrane Risk of Bias tool (RoB 2.0). Sixteen trials were included for qualitative analysis. More than half of the trials do not adequately report the characteristics that make up the exercise program. The maximal strength was the most frequently monitored manifestation of strength, evaluated mainly as one-repetition maximum (1RM). Resistance training was performed on strength-training machines, twice a week, using a load between 50% and 80% of 1RM. The trials reported significant improvement in muscle strength, fatigue, pain, quality of life, and minor changes in aerobic capacity.

## 1. Introduction

The health benefits of physical exercise (PE) have been extensively described, highlighting its positive influence on physical, psychological, cognitive, and social condition, as well as its role in disease prevention and treatment, and improvement of quality of life (QoL) [1]. In relation to breast cancer (BC), PE may be an important element in the reduction of risk factors, mortality, and relapse [2,3,4]. It diminishes the sequelae produced by the treatments against BC, such as the loss of joint mobility and muscle strength, pain, fatigue, anxiety, and depression [5,6]. PE is usually considered a component of the global treatment of disease [7]; it does not cause adverse events and improves the overall condition of breast cancer survivors (BCS) [7,8,9].

Muscle contraction generates a tension in the muscle, which applied by osteoarticular levers, is able to oppose or overcome an imposed load. This capacity is generically referred to as muscular strength. However, the different outcomes or performances of muscle strength are diverse and are given different names, some of which are used in this review (Table 1).

Resistance training (RT) alone is less commonly used than aerobic training (AT) or combined training (AT + RT), perhaps due to many patients’ false belief that exercising with their arms can produce lymphedema [10]. However, the studies carried out so far have ruled out such an idea, and have identified valuable benefits of RT for the musculoskeletal system, joint mobility, fatigue, depression, self-image, and QoL [11,12,13]. This is why muscle strength exercises are increasingly being included in training programs for breast cancer survivors [2,14]. In the literature, a broad range of RT programs developed for BCS can be found, but there is still no consensus on the frequency, intensity, mode, or timing of their prescription [4,15].

In view of such benefits, and in order to make an appropriate prescription for RT, we consider the current knowledge of aspects such as the manifestations of muscle strength that have been used (maximal strength, power, and resistance endurance); evaluation methods (one-repetition maximum, hypothetical maximum force test, use of encoders, body weight, or other methods); frequency of sessions; and intensity of training loads; as well as the results obtained.

The aim of this systematic review was to identify the characteristics of muscle strength evaluation (manifestations of strength, exercises, and muscle groups), the training programs (supervision, resistance type, duration, sets, repetitions, intensity of exercise, sessions per week, and muscle groups or exercises), the secondary objective, the following variable outcomes that were analyzed in each study, the general result obtained, and the safety of the RT in the studies published from 1 January 1990 to 6 December 2019, that have used RT in BCS, before, during, or after treatment.

## 2. Methods

### 2.1. Search Strategy

A systematic review of the literature was performed using PubMed, Medline, Science Direct, the Cochrane Breast Cancer Specialised Register of the Cochrane Library, the Physiotherapy Evidence Database (PEDro), and Scopus, with the aim of identifying all published studies on RT and BCS from 1 January 1990 to 6 December 2019. In addition, potential articles were searched in the reference lists of identified trials and reviews. Two key terms, ‘weight training’ and ‘breast cancer’, were used to generate an exhaustive list of keywords. Table 2 shows the full search strategy.

### 2.2. Study Selection

Two reviewers independently classified and verified the extracted data to present a descriptive summary of the important features of each study. Any disagreement between those reviewers was resolved through discussion with a third reviewer. The titles and abstracts of the remaining articles were examined for eligibility, and those selected were read in full to determine their inclusion according to the pre-defined criteria (Table 3). The excluded studies were those that prescribed a nonconventional RT such as tai chi, Pilates, yoga, Nordic walking, or aquatic exercise. Studies which used different interventions to the RT (e.g., stretching sessions, dietary modification) were excluded. When the sample included patients with more than one type of cancer, the article was included only if the results of RT in BCS were presented independently. The studies that were available only as abstracts were also excluded. We performed a systematic review of the literature using the Preferred Reporting Items for Systematic Reviews and Meta-Analyses (PRISMA) guidelines [16,17].

### 2.3. Data Extraction and Quality Assessment

To respond to the main objective of our study, a specific form was developed to extract the relevant information from each of the studies analyzed: the objective, the time of the intervention, the characteristics of the population (e.g., sample size, age, treatment), the study groups, the characteristics of the muscle strength assessment (strength manifestations, exercises, and muscle groups), the RT program (supervision, type of resistance, duration, series, repetitions, exercise intensity, sessions per week, and muscle groups or exercises). To attend to our secondary objective, in each study we analyzed the general result obtained and the safety of the RT.

Two authors independently assessed the risk of bias. In the case of disagreement, the subject was discussed with another author. The risk of bias was assessed using the Cochrane risk-of-bias tool for randomized trials (RoB 2.0) [18,19], which evaluates the risk of bias in five domains; the randomization process, deviations from intended interventions, missing outcome data, measurement of the outcome, and selection of the reported result. A study is considered to be at a “low risk of bias” if all five domains have been judged to be at a low risk of bias. A study is considered to have “some concerns” if it has been judged to raise some concerns in at least one domain. A study is considered to be at a “high risk of bias” overall if it is judged to be at a high risk of bias in at least one domain.

## 3. Results

### 3.1. Study Selection

With the use of the six databases, 133 trials were chosen to be read in full, since they presented information about the developed RT programs. From these 133 studies, 86 corresponded to clinical trials, 31 to systematic reviews, and 16 to systematic reviews and meta-analyses. Finally, 41 articles that met the selection criteria were selected (Figure 1). These 41 articles were derived from a total of 16 trials (Table 4) [7,8,10,20,21,22,23,24,25,26,27,28,29,30,31,32,33,34,35,36,37,38,39,40,41,42,43,44,45,46,47,48,49,50,51,52,53,54,55,56]. The BEATE study [26,27] and the BEST study [8,28,29,30] were treated as a single trial in the present review because they shared the same training program. The PAL [46,47,48,49,50,51,52,53,54] study and that of Buchan et al. (2016) [55] also shared the same training program.

### 3.2. Characteristics of the Selected Studies

The scores of the included trials on the PEDro scale are presented in Table 5. Although the first item does not contribute to the total score because it is related to the external validity [15], all the included trials met the external validity item by clarifying the eligibility criteria. All the included trials met the random allocation criteria, reporting between-group differences and point estimates and variability, which contributed to the total score. Only one trial did not report similar groups at baseline [39], five did not use a valid allocation concealment method [23,24,37,40,45], three trials were blinded to the participants [7,26,40], five trials had blinded therapists [26,31,33,38,56], and eight trials had blinded assessors [7,20,25,31,33,35,38,46]. Six trials had >15% loss to follow-up [20,25,31,35,38,46], and four were not analyzed by intention to treat [31,35,38,39]. The risk of bias analysis showed a low risk of bias in five articles, a moderate risk of bias in nine articles, and a high risk of bias in three articles (Figure 2).

The total sample size of the studies included in the review was 1835 participants. The sample sizes of the individual studies varied between 23 and 295 participants. Seven experimental populations utilized samples of more than 100 BCS [20,24,25,28,32,37,46]. None of the articles reported the inclusion of male participants. The mean age ranged from 47 to 64 years old; there were only two experimental populations in which the mean age of participants was below 50 years old [20,24]. The BC stage of participants varied between stage 0 and stage III. Stage IV was reported in only one experimental population (Table 6) [26].

The most common aim of the selected articles was to identify the effects of RT on QoL (36.3%) [7,8,9,20,27,33,36,38,39,48,54,55,56], followed by identifying the effects of RT on body composition (26.6%) [20,22,23,24,25,35,40,52,53], and on lymphedema (26.6%) [10,20,25,32,35,47,49,50,55,56]. None of the trials investigated a possible preventive role of RT on the development of BC. In the trials in this review, there were two to three study groups that included only BCS, and none of the trials established a comparison with a BC-free population. Of the selected studies, 37.5% compared trained groups with groups that did not follow any training protocol and that continued their regular care [7,25,32,37,46,56]. Another 43.7% used more than one group including a different activity (AT, RT + supplement, relaxation, or usual care) and compared the results with those of RT [20,23,24,31,38,40,45]. Only one of the articles represented a study comparing different load intensities (high and low) [56].

### 3.3. Manifestations of Muscle Strength and Evaluation Methods

Maximal strength was evaluated in all the trials. The evaluation method and tested limbs were unspecified in one of the 16 trials [25]. The test was performed on the upper and lower extremity, except for one study that evaluated the muscle strength of the upper body (UB) using resistance-endurance [7]. Three trials evaluated resistance-endurance, two of them measured it in the UB [7,38] and one in the lower body (LB) and UB [56].

The most commonly adopted methods of evaluating maximal strength were the one-repetition maximum (1RM), the multiple repetition test (MRT), and the hypothetical maximum force test (h1RM). Eight trials used 1RM to prescribe the RT [7,23,24,35,40,45,54,56], four used the MRT (4RM–8RM) [20,32,38,55], three used the h1RM [26,31,39], two used maximal grip strength [32,56], and one trial did not report any evaluation method [25]. Some studies mentioned the muscle group that was evaluated [26,37] and others specified the type of movement or exercise used [20,23,24,35,38,39,40,55]. Six exercises were used to test the muscle strength of the UB. The most commonly used exercises were the bench/chest press [7,20,31,35,38,39,40,45,46,47,56] and the seated row [23,24,38,56]. The most common exercise to evaluate the muscle strength of the LB was the knee extension [7,20,23,24,26,31,35,37,38,39,40,45,46,56]. The evaluation methods mentioned were used during and after BC treatment (Table 7).

### 3.4. Exercise Prescription

More than half of the trials developed a supervised training program [7,20,25,26,31,39,40,45,56]. The duration of the training programs varied between 12 and 96 weeks, with 12 weeks being the most frequent value [7,26,31,38,55,56]. Only one trial did not mention the exact duration of the program, because it depended on the duration of chemotherapy [20]. The frequency ranged from one to four times per week, with two sessions being the most frequent interval [26,31,32,35,37,40,45,46,56]. Each session lasted between 20 and 90 min, with 60 min being the most frequent duration [7,26,32,35,56]. Six trials did not mention the duration of the training session [20,23,38,39,40,45].

Regarding the intensity of the RT of the UB, this variable was not specified in four trials [23,24,35,37]; those trials utilized a level of intensity based on the participants’ tolerance. In the other trials, variable intensity was employed, with moderate intensity being the most frequent. Low intensity, defined as lower than 50% of 1RM or equal to 20–25 RM, was used in three trials [25,32,47]. Eleven trials utilized a moderate intensity, between 50% and 80% of 1RM or equal to 8–19 RM [7,20,26,31,32,35,39,40,45,46,56]. Five trials used a high intensity, greater than 80% of 1RM or equal to 5–7 RM [20,30,35,46,56]. One trial used the rated perceived exertion (RPE) scale [38]. The number of sets varied between one and four, but most trials (over 80%) used two or three sets per exercise [7,20,23,24,26,32,35,37,39,40,45,46,56]. One trial did not report the number of sets [25]. The number of repetitions varied between 8 and 20; 50% used 8–12 repetitions (Table 8) [7,20,23,26,35,37,39,40]. The type of resistance used included strength-training machines [7,26,31,35,37,39,40,45,46], resistance bands [23,24,38], dumbbells, and self-loading [7,24,32,35,37,46]. In three studies, the type of resistance used was not mentioned (Table 9) [20,25,56]. Nine trials focused on the training of the upper and lower limbs [20,23,24,27,31,35,45,46,56]. Two trials made no mention of the trained muscular group [25,32]. The other trials aimed to develop muscle strength in the upper and lower limbs and trunk [7,35,37,38,39,40]. The most used exercise was the seated row [7,20,24,26,38,39,40,45,46,56], and the chest press [7,20,31,35,38,39,40,45,46,56]. The most common exercise of the LB was the leg press [7,20,24,26,37,38,39,40,45,46,56] and the leg extension [7,20,23,26,31,37,40,45,46,56]. 

### 3.5. Results and Safety of Resistance Training

Eight trials evaluated aspects of the body composition, six trials reported improvements [20,23,35,40,46,51,53], and two did not find changes in any of the evaluated variables (muscle mass, mineral bone density, fat mass, or body mass index). Twelve trials reported significant improvement in muscle strength [7,20,23,24,26,35,37,38,40,45,46,56], whereas the others did not provide information [20,25,31,32,39]. Six trials assessed aerobic capacity, with two reporting significant improvement [24,55] and two reporting minor changes [20,23,38,39]. There was a significant improvement in aspects related to QoL [7,20,26,33,35,39,46,56], self-perception [20,38,46], balance [37], joint range of motion [56], and fatigue and pain [7,20,25,26,38] in all evaluated cases [7,20,26,33,35,39,46,56]. Only one of the three trials evaluating depression reported a significant improvement [38]. None of the trials reported changes in the participants’ physical activity habits after the conclusion of the training program. Regardless of when the resistance training took place (during/after the BC treatment), QoL, self-perception, pain, fatigue, body composition, and muscle strength showed significant improvements. Most studies did not report changes in aerobic capacity or lymphedema during or after treatment (Table 10).

According to the reviewed studies, safety was not affected by the physical exercises carried out, the type of resistance used (free weight, resistance bands, dumbbells, strength-training machines), or the load intensity. Eleven trials did not report any adverse effects of RT [7,20,24,25,26,32,35,37,45,54,56], and the other six studies did not report about training safety [7,23,31,37,38,39,40]. In relation to the effect of PE on lymphedema in BCS, none of the six trials that evaluated this relation found an increase in lymphedema [20,25,26,32,35,46,56].

Detailed data from the studies are available as Appendix A: Characteristics of controlled trials reviewed, Appendix A: Outcomes, measuring tools, and main findings, Appendix A: Exercise prescription.

## 4. Discussion

Of the 133 publications initially considered, 47 were reviews and meta-analyses with different RT approaches in BCS, which illustrates the interest in unifying and corroborating the effects of this type of training. Nevertheless, it is important to take into account that more than half of the publications found were derived from only 16 trials. This small reference pool reveals the need for more research in this area. Another limitation concerned the samples used in the trials, most of which included fewer than 80 participants; only two of the trials had large sample sizes (242 and 295 BCS, respectively) [20,46]. It is necessary to unify the designs of the interventions with large samples that provide comparable information and more valid conclusions.

A few of the systematic reviews focused on RT in BCS populations. Most reviews described the general characteristics of the trials, training program, and obtained results in contrast with other types of training, especially with cancer survivors having, or at risk of developing, lymphedema related to BC; however, these descriptions were generic and not explicit enough regarding the resistance evaluation method, exercise intensity, progress, the type of resistance and exercise used during evaluation, and the training program. This lack of detail constitutes a limitation when comparing the results, the possibility of implementing training programs in other contexts, and at the same time, unifying the recommendations related to RT for BCS.

One of the greatest fears of BCS in relation to RT is the worsening of symptoms, or the exacerbation of lymphedema, which constitutes one of the most discussed topics in the selected literature [11,13,14,15,57]. The results of the studies on RT suggest that if there is control and progression in the training, considering the individuality of the participants, then there will be no increased risk or worsening of the symptoms or severity of lymphedema [13,15], regardless of the evaluation protocol, training program, or timing of intervention (during or after treatment). It can be said that this type of training is safe, and that contrary to the general precept, lymphedema symptoms and exacerbation are not increased by this form of training [20,25,26,32,35,46,56,57].

Training also influences the patients’ psychological condition and quality of life during and after cancer treatment [11]. Variables such as body composition and aerobic capacity did not always show significant changes, perhaps due to the intensity, frequency, or type of training used. The effects of RT on muscle strength and joint motion were not often considered as research objectives, despite the fact that they were evaluated in most the trials for the prescription of PE, and they were two of the aspects that were impaired after BC surgery and were related to QoL. In the two studies that evaluated joint motion [38,56], improvement was observed in the range of motion, without any adverse effects, among women who had concluded chemotherapy and radiation therapy. Similarly, muscle strength improved significantly [7,20,23,24,26,35,37,38,40,45,46,56], sometimes exceeding the amount of muscle strength lost after the surgery [14]. Muscle strength is important, in that it reduces musculoskeletal injuries [15]. Apparently, there was a significant gain of muscle strength regardless of the duration of the training program, training intensity (high, low, moderate), or timing of intervention [20,35,37,40,47,56]. Some studies mentioned that training during treatment led to better and faster effects on mobility than late interventions; however, further evidence is required. Determining which training regimens are most effective in improving muscle strength, range of motion, conservation of bone structure, and reduction of fat mass could be a potentially interesting research direction. Some studies have reported isolated data, but there is a need for more evidence.

Considering the timing of the development of the training program, it is suggested that RT prior to the administration of BC treatment would help patients address the damage and alterations caused by the treatment, such as the loss of muscle mass, muscle strength, and mobility, thus also affecting health-related QoL, which was better in people who were physically active before diagnosis than in those who were sedentary [14].

In light of the published studies, professionals who care for the health of breast cancer survivors can inform their patients that RT is safe and can provide important benefits. Although the quality of the description of the RT programs carried out needs to be further improved, there are sufficient studies that can guide professionals in prescribing this type of exercise, ensuring safety in training programs, and providing guidance as to their frequency, load, number of series and repetitions, and methods of carrying them out.

## 5. Conclusions

Most studies used the evaluation of maximal strength to develop training programs, none of the studies performed an evaluation of muscle power, and only three studies evaluated resistance-endurance.

RT in BCS is typically performed on strength training machines, twice a week, using a load between 50% and 80% of 1RM, with sessions of 60 min and with two or three sets of 8 to 12 repetitions for each muscle group worked.

The measured outcomes of intervention with this type of training mostly focused on the effect on QoL, followed by the effects on lymphedema, fatigue, and body composition.

In view of these studies, it can be emphasized that RT is safe, that it does not adversely affect the development or worsening of lymphedema, and that it helps improve the QoL of these patients.

RT can be considered an additional treatment with which to supplement adjuvant and rehabilitation therapy for BCS. It is necessary that trials describe with sufficient precision the manifestation of muscle strength studied, the methods for evaluating it, and the method of individualizing the training load, which would allow these studies to be replicated and compared.

## Figures and Tables

**Figure 1 ijerph-17-06511-f001:**
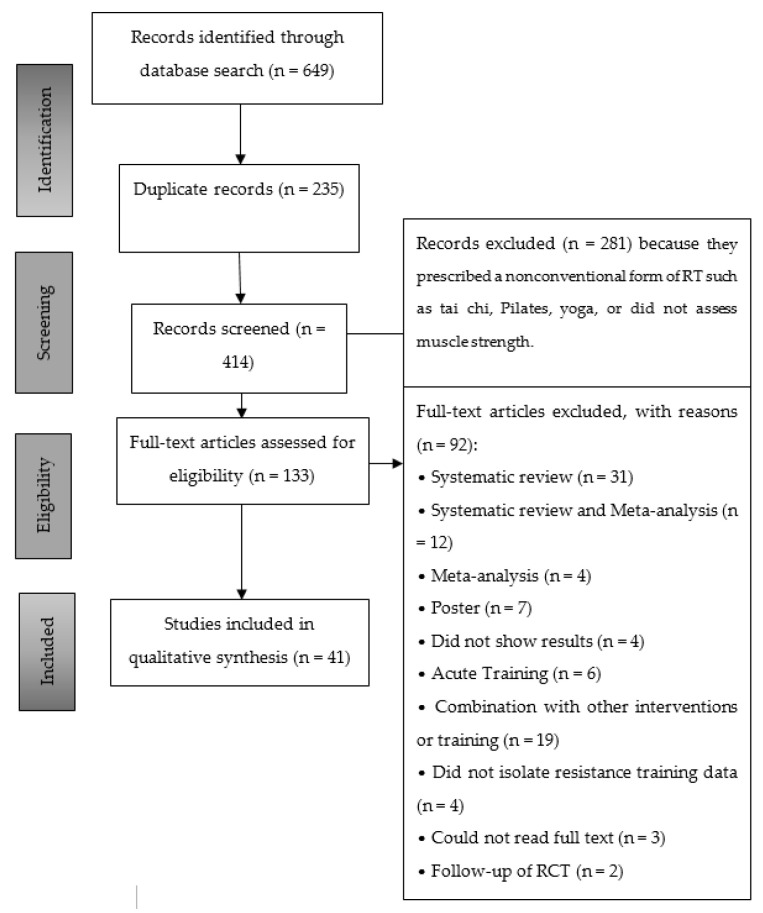
Flow chart for systematic review methodology as per Preferred Reporting Items for Systematic Reviews and Meta-Analyses (PRISMA) guidelines. RT = resistance training; RCT = randomized controlled trial.

**Figure 2 ijerph-17-06511-f002:**
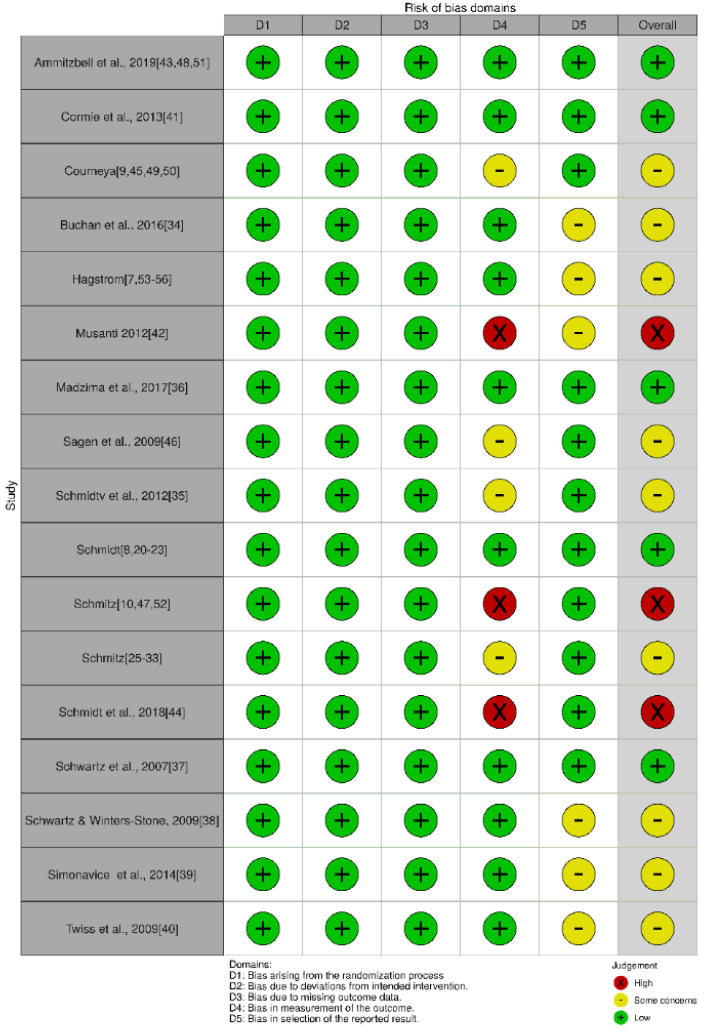
Risk of bias assessment of the randomized trials.

**Table 1 ijerph-17-06511-t001:** Definitions of terms used in this review.

Terms	Definitions
Maximum dynamic force	The maximum load you are able to mobilize once (one repetition maximum, 1RM).
Muscle power	The result of the product of the load imposed by the speed at which it moves in a movement characterized by running in a very short period of time, short and high-intensity.
Resistance force (or resistance endurance)	The maximum number of times you are able to mobilize a load.
Resistance training	The generic name used to refer to chronic exercise aimed at maintaining or improving neuromuscular performance.

**Table 2 ijerph-17-06511-t002:** Search Terms.

Search Mode	Advanced
Database coverage date	From 1 January 1990 to 6 December 2019
Search term 1	OR weight train * OR weight lift * OR resistance train * OR resistance exercise * OR progressive resisted exercise * OR weight-bearing exercise * OR strength exercise * OR strength train * OR strengthening program * OR exercise training * OR exercise program * OR physical activity * OR physical exercise * OR rehabilitation *
Search term 2	breast cancer * OR breast neoplasm * OR breast carcinoma * OR breast tumor * OR mammary neoplasm * OR mammary malignant * OR mammary carcinoma * OR mastectomy * OR lymph node excision * OR lymphedema *
Search	Search term 1 AND Search term 2

* Truncation symbol to retrieve terms with a common root.

**Table 3 ijerph-17-06511-t003:** Inclusion criteria.

Design	Randomized Trials
Population	Breast cancer survivors
Intervention	Resistance training
Outcomes	Muscle strength evaluation Resistance training characteristics Resistance training results
Comparison	Control group (without the disease, survivors of breast cancer who developed other types of training, breast cancer survivors that remained sedentary) Pre and post-training

The quality of the selected studies was assessed using the PEDro scale [16,17]. This review includes only randomized trials that applied resistance training (RT) programs in breast cancer survivors (BCS) during or after treatment (surgery, axillary node dissection, radiotherapy, chemotherapy, or hormonal therapy) published from 1 January 1990 to 6 December 2019.

**Table 4 ijerph-17-06511-t004:** Studies included in the review.

No.	TRIALS	ARTICLES
**1**	START: Supervised Trial of Aerobic Versus Resistance Training	Courneya et al., 2007 [20]-Courneya et al., 2007 [21] Courneya et al., 2014 [9]-Adams et al., 2016 [22]
**2**	Schwartz et al., 2007 [23]	
**3**	Schwartz & Winters-Stone, 2009 [24]	
**4**	Sagen et al., 2009 [25]	
**5**	BEATE: Exercise and relaxation as therapy against fatigue BEST: Exercise and relaxation for breast cancer patients during radiotherapy	Schmidt et al., 2013 [26]-Schmidt et al., 2015 [27]
		Potthoff et al., 2013 [28]-Steindorf et al., 2014 [8] Schmidt et al., 2016 [29]-Wiskemann et al., 2017 [30]
**6**	Schmidt et al., 2018 [31]	
**7**	Ammitzbøll and colleagues	Ammitzbøll et al., 2019 [32]-Ammitzbøll et al., 2019 [33] Ammitzbøll et al., 2019 [34]
**8**	WTBS: Weight Training for Breast Cancer Survivors	Schmitz et al., 2005 [35]-Ohira et al., 2006 [36] Ahmed et al., 2006 [10]
**9**	Twiss et al., 2009 [37]	
**10**	Musanti, 2012 [38]	
**11**	Schmidt et al., 2012 [39]	
**12**	Simonavice et al., 2014 [40]	
**13**	Hagstrom and colleagues	Hagstrom et al., 2015 [7]-Hagstrom et al., 2016 [41] Hagstrom, A. D., Shorter, K. A., & Marshall, P. W. 2019 [42]-Hagstrom, A., & Denham, J. 2018 [43] Hagstrom, A. D., & Denham, J. 2018 [44]
**14**	Madzima et al., 2017 [45]	
**15**	PAL: Physical Activity and Lymphoedema	Schmitz et al., 2009 [46]-Schmitz et al., 2009 [47] Speck et al., 2010 [48]-Schmitz et al., 2010 [49] Hayes et al., 2011 [50]-Brown et al., 2012 [51] Winters-Stone et al., 2014 [52] Brown & Schmitz et al., 2015 [53] Brown & Schmitz et al., 2015 [54]
	Buchan et al., 2016 [55]	
**16**	Cormie et al., 2013 [56]	

**Table 5 ijerph-17-06511-t005:** Physiotherapy Evidence Database (PEDro) scores of the included trials (n = 16).

Study	Random Allocation	Concealed Allocation	Groups Similar at Baseline	Participant Blinding	Therapist Blinding	Assessor Blinding	<15% Dropouts	Intention-to-Treat Analysis	Between-Group Difference Reported	Point Estimate and Variability Reported	Total (0 to 10)
Ammitzbøll et al., 2019 [32,33,34]	Y	Y	Y	N	Y	Y	Y	Y	Y	Y	9
Cormie et al., 2013 [56]	Y	Y	Y	N	Y	N	Y	Y	Y	Y	8
Courneya [9,20,21,22]	Y	Y	Y	N	N	Y	N	Y	Y	Y	7
Buchan et al., 2016 [55]	Y	Y	Y	N	N	N	Y	Y	Y	Y	7
Hagstrom [7,41,42,43,44]	Y	Y	Y	Y	N	Y	Y	Y	Y	Y	9
Musanti 2012 [38]	Y	Y	Y	N	Y	Y	N	N	Y	Y	7
Madzima et al., 2017 [45]	Y	N	Y	N	N	N	Y	Y	Y	Y	6
Sagen et al., 2009 [25]	Y	Y	Y	N	N	Y	N	Y	Y	Y	7
Schmidtv et al., 2012 [39]	Y	Y	N	N	N	N	Y	N	Y	Y	5
Schmidt [8,26,27,28,29]	Y	Y	Y	Y	Y	N	Y	Y	Y	Y	9
Schmitz [10,35,36]	Y	Y	Y	N	N	Y	N	N	Y	Y	6
Schmitz [46,47,48,49,50,51,52,53,54]	Y	Y	Y	N	N	Y	N	Y	Y	Y	7
Schmidt et al., 2018 [31]	Y	Y	Y	N	Y	Y	N	N	Y	Y	7
Schwartz et al., 2007 [23]	Y	N	Y	N	N	N	Y	Y	Y	Y	6
Schwartz & Winters-Stone, 2009 [24]	Y	N	Y	N	N	N	Y	Y	Y	Y	6
Simonavice et al., 2014 [40]	Y	N	Y	Y	N	N	Y	Y	Y	Y	7
Twiss et al., 2009 [37]	Y	N	Y	N	N	N	Y	Y	Y	Y	6

Y = yes, N = no.

**Table 6 ijerph-17-06511-t006:** Characteristics of controlled trials reviewed.

Trial/Author	N	Age	Stage of Cancer	Treatment	Control Groups
**During treatment**					
START [9,20,21,22]	242	49.2	I–IIIA	M-C	AT-UC
Schwartz et al., 2007 [23]	66	50.1 ± 8.7	I–III	C-R	AT-UC
Schwartz & Winters-Stone, 2009 [24]	101	47 ± 9.4	I–III	C	AT-UC
Sagen et al., 2009 [25]	204	55 ± 10	I–III	M/AND-C-R-HT	UC
BEATE [26,27]	95	52.7 ± 10	I–IV	M/L-C	PMR
BEST [8,28,29,30]	155	55.8 ± 9.1	0–III	NC-M/L-R-HT	PMR
Buchan et al., 2016 [55]	40	56	NR	S-C-R-HT	AT
Schmidt et al., 2018 [31]	67	54	NR	S-C	AT-UC
Ammitzbøll et al., 2019 [32,33,34]	158	52	I–III	S/AND-C-HT	UC
**Post-treatment**					
WTBS [10,35,36]	79	53.3 ± 8.7	I–III	AND-C-R	UC
Twiss et al., 2009 [37]	110	58.7 ± 7.5	0–II	S-C-R	UC
Musanti, 2012 [38]	42	50.5	I–IIIB	C-R	AT-CT-F
Schmidt et al., 2012 [39]	33	58 ± 8.4	I–III	M/L-C-R	CGE
Simonavice et al., 2014 [40]	23	64 ± 5	0–III	S-C-R-HT	RT+DP
Hagstrom and colleagues [7,41,42,43,44]	39	51.9 ± 8.8	I–IIA	S-C-R-HT	UC
Cormie et al., 2013 [56]	62	57 ± 10	0–III	S/AND-C-R-HT	UC
PAL [46,47,48,49,50,51,52,53,54]	295	55.3 ± 8.5	I–III	S/AND-C.HT	UC
Madzima et al., 2017 [45]	33	59 ± 9	0–III	S-C-R-HT	RT + PRO

START = Supervised Trial of Aerobic Versus Resistance Training; BEATE = exercise and relaxation as therapy against fatigue; BEST = exercise and relaxation for breast cancer patients during radiotherapy; WTBS = Weight Training for Breast Cancer Survivors; PAL = Physical Activity and Lymphoedema; UC = usual care; R = radiotherapy; C = chemotherapy; NC = Neoadjuvant chemotherapy; HT = hormonal therapy; S = surgery; M = mastectomy; L = lumpectomy; AND = axillary node dissection; AT = aerobic training; PMR = Progressive muscle relaxation; CGE = conventional gymnastics exercise; F = Flexibility; CT = combined training; RT + DP = resistance training + dried plum; RT + PRO = resistance training+ protein supplementation.

**Table 7 ijerph-17-06511-t007:** Manifestations, methods and muscular groups evaluated.

Manifestation of Muscle Strength	Evaluation Method		Movements/Muscular Groups	
	UB	LB	UB	LB
**During Treatment**				
Maximal strength	NR [25]		Chest press [9,20,21,31,55]	Leg extension [9,20,21,23,24,31]
	1RM [23,24]		Seated row [23,24]	Knee extensors [8,26,27,28,29]
	MRT (4-8 RM) [9,20,21,22,32,33,34,55]		Shoulder press [23,24,31]	Knee flexors [8,26,27,28,29]
	h1RM - Brzycki-Method [8,26,27,28,29,30,31]		Shoulder rotators [8,26,27,28,29]	Leg curl [31]
	Maximal grip strength [32,33,34]	-	Upper arm curl and extensors [31]	Squat [31]
	Isometric strength protocol [32,33,34]	-	Latissimus pull down - Sit-ups [31]	Rowing [31]
			Shoulder abductors, adductors,	Leg press [32,33,34]
			flexors, and extensors [32,33,34]	
			Elbow flexion and extension [32,33,34]	
**Post-treatment**				
Maximal strength	1RM [10,35,36,40,45,46,47,48,49,50,51,52,53,54,56]	1RM [7,10,35,36,40,41,45,46,47,48,49,50,51,52,53,54,56]	Chest press [7,10,35,36,38,39,40,41,45,46,47,48,49,50,51,52,53,54,56]	Leg extension [10,35,36,38,39,40,45]
	Maximal grip strength [56]			
	MRT (6 RM) [38]		Seated row [38,56]	Leg press [7,41,46,47,48,49,50,51,52,53,54,56]
	BVSE [37]		Shoulder press [39]	Leg curl [39]
	h1RM [39]		Latissimus pull down [39]	Squat [39]
	Unilateral isometric strength protocol [7,41]	-	Sit ups [38,39]	Rowing [39]
Strength-Endurance	Curl-up test [38]	-	Upper arm curl and extensors [39]	Hip [37]
	YMCA Bench Press Endurance Test [38]	-	Wrist [37]	Knee extensors [37]
	Repetition maximum test [56]	-	Handgrip Strength [56]	Knee flexor [37]

1RM = one repetition maximum; MRT = multiple repetitions tests; h1RM = hypothetical maximum force test; BVSE = Biodex velocity spectrum evaluation; UB = upper body; LB = lower body; NR = Not reported.

**Table 8 ijerph-17-06511-t008:** Main characteristics of Exercise Prescription.

Trial Duration (Wk.)	Exercise Intensity	Increment	Sets	Repetitions	Sessions/Wk.	Session Duration/Min
12 [7,8,26,27,28,29,31,38,41,45,55,56]	NR [10,23,24,35,36,37]	NR [7,8,24,26,27,28,29,41]	NR [25]	8–10 [10,23,35,36]	1 [39]	NR [9,20,21,23,38,39,40,45]
17 [9,20,21]	<50% 1RM [32,33,34] UB: 1-pound weights [46,47,48,49,50,51,52,53,54] 0.5 kg. [25]	10% → >12 reps/set [9,20,21,40] 5%–10% → 2 Sess. [56]	1 [10,31,35,36,38,56]	8–12 [7,8,9,20,21,26,27,28,29,37,39,40,41]	2 [8,10,26,27,28,29,31,32,33,34,35,36,37,40,45,46,47,48,49,50,51,52,53,54,55,56]	20–30 [24]
20–30 [10,23,25,32,33,34,35,36,39,40]	50%–80% 1RM [7,8,9,20,21,26,27,28,29,31,39,40,41,45,56]	Modifying starting grip position [23] Decreasing 5RM x module [32,33,34]	2 [9,10,20,21,23,24,35,36,37,39,40,46,47,48,49,50,51,52,53,54,55,56]	10–12 [10,35,36,38,45,46,47,48,49,50,51,52,53,54,55]	3 [7,9,20,21,38,41]	>30–45 [37]
48 [10,24,35,36,46,47,48,49,50,51,52,53,54]	>85% 1RM [10,35,36,46,47,48,49,50,51,52,53,54,55]	Tolerance [10,25,35,36,37,39] RPE = ≤3 [31,38]	3 [7,8,10,24,26,27,28,29,32,33,34,35,36,41,45,46,47,48,49,50,51,52,53,54,55,56]	11–18/20 [24,25,31,32,33,34]	4 [23,24]	50–60 [8,10,26,27,28,29,35,36]
96 [37]	RPE of 3–5 [38]	UB = 1/2 pound → 2 Sess. [46,47,48,49,50,51,52,53,54,55]	4 [56]	20–15 RM 10-6 RM [56]		>60–90 [46,47,48,49,50,51,52,53,54]
-	-	LB = Smallest possible increment [46,47,48,49,50,51,52,53,54,55] 1.81 kg→ >10 reps/third set [45]	-	-	-	-

PP = Part of the program; NR = not reported; BW = body weight; 1RM = one-repetition maximum; UB = upper body strength; LB = lower body strength; RPE = rating of perceived exertion; Reps = repetitions; Sess. = sessions; Wk. = week; HL = high load; LL = low load.

**Table 9 ijerph-17-06511-t009:** Muscular Groups Exercised.

Supervised Training	Resistance	Movements/Muscular Groups	
		UB	LB
YES [7,8,9,20,21,25,26,27,28,29,31,39,40,41,45,56]	NR [9,20,21,22,25,56]	Chest press [7,9,10,20,21,31,35,36,38,39,40,41,45,46,47,48,49,50,51,52,53,54,55,56]	Leg extension [7,8,9,20,21,23,26,27,28,29,31,37,40,41,45,46,47,48,49,50,51,52,53,54,55,56]
NO [23,24,38]	Bands [23,24,38]	Seated row [7,8,9,20,21,24,26,27,28,29,38,39,40,41,45,46,47,48,49,50,51,52,53,54,55,56]	Leg press [7,8,9,20,21,24,26,27,28,29,37,38,39,40,41,45,46,47,48,49,50,51,52,53,54,55,56]
PP [10,32,33,34,35,36,37,46,47,48,49,50,51,52,53,54,55]	BW–Dumbbells [7,10,24,32,33,34,35,36,37,41,46,47,48,49,50,51,52,53,54,55]	Shoulder press [23,24,31,38,39,45,56]	Leg curl [7,8,9,20,21,26,27,28,29,31,37,40,41,45,46,47,48,49,50,51,52,53,54,55]
-	Machines [7,8,10,26,27,28,29,30,31,35,36,37,39,40,41,45,46,47,48,49,50,51,52,53,54,55]	Lateral, front, and up raise shoulder [46,47,48,49,50,51,52,53,54,55,56]	Squat [31,38,39,56]
-	-	Latissimus pull down [7,8,26,27,28,29,31,37,38,39,41,56]	Calf raises [9,20,21,37]
-	-	Triceps extension [9,20,21,31,37,38,39,40,56]	Lunge [37,56]
-	-	Triceps pushdown [40,45,46,47,48,49,50,51,52,53,54]	Hip flexion [37,38,39]
^-^	-	Bicep curl [9,20,21,31,37,38,39,40,45,46,47,48,49,50,51,52,53,54,55,56]	Hip extension [37,38,39]
-	-	Wrist curl [37,56]	Lower back hyperextension [7,40,41,45]
-	-	One-arm row–barbell bent [7,41,46,47,48,49,50,51,52,53,54,55]	Buttocks, thighs, and legs [10,35,36]
-	-	Butterfly and butterfly reverse [8,26,27,28,29]	Rowing [31]
-	-	Upward row–Push-ups/push-ups on knees–Side hip raise [37]	NR [25,32,33,34]
-	-	Sit-ups [7,31,37,38,39,40,41,45]	-
-	-	Prone hold [7,41] Seated [23]	-
-	-	Ball-gripping–Wrist extension [37]	-
-	-	Shoulder flexion; shoulder extension [38]	-
-	-	Back extension [7,37,41,46,47,48,49,50,51,52,53,54,55] back [10,35,36]	-
-	-	Shoulder rotators [8,26,27,28,29] shoulders [10,35,36]	-
-	-	NR [25,32,33,34]	-
-	-	-	-

PP = part of the program; NR = not reported; BW = body weight; UB = upper body strength; LB = lower body strength.

**Table 10 ijerph-17-06511-t010:** Results and safety of resistance training.

Result	Training Safety
**During Treatment**	
↑Muscle strength [8,9,20,21,23,24,26,27,28,29,30,55]	No adverse events [8,9,20,21,22,24,25,26,27,28,29,30,32,33,34,55]
↑ Aerobic capacity [24,55]	NR [23,31]
↔ Aerobic capacity [8,9,20,21,26,27,28,29]	-
(↑) Aerobic capacity [23]	-
↓ Fatigue [8,9,20,21,26,27,28,29]	-
↔ Bodyweight [24]	-
↔ Body composition [24]	-
Attenuates the decrease in BMD [23]	-
↑ LBM [9,20,21]	-
↔ Upper Limbs Volume [9,20,21,25,32,33,34,55]	-
↓ Sarcopenia and Dynapenia [22]	-
↑ QoL [8,9,20,21,22,26,27,28,29,33,55]	-
↑ Self-perceptions [9,20,21]	-
↓ Anxiety [9,20,21]	-
↔ Depression [8,26,27,28,29]	-
↓ Pain [8,25,26,27,28,29]	-
Stronger effects on DFS, OS, DDFS, RFI [9,20,21]	-
(↑) Cognitive performance [8,26,27,28,29]	-
↓ IL-6, IL-6/IL-1ra [8,26,27,28,29]	-
Not suppress cellular immunity [31]	-
**Post Treatment**	
↑Muscle strength [7,10,35,36,37,38,40,41,42,45,46,47,48,49,50,51,52,53,54,56]	↔ Incidence of fractures or falls [37,46,47,48,49,50,51,52,53,54]
↑Muscle endurance [56]	NR [7,38,39,40,41]
↔ EMG [42]	No adverse events [45,46,47,48,49,50,51,52,53,54,56]
(↑) Aerobic capacity [39]	↓ Number and severity of symptoms [46,47,48,49,50,51,52,53,54]
↓ Fatigue [7,38,41]	-
↑ Perceived exertion [39]	-
(↑) ROM [38]	-
↑ ROM [56]	-
↔ BMI [10,35,36,40,46,47,48,49,50,51,52,53,54]	-
(↓) BMI [39]	-
↔ Body weight [10,35,36,46,47,48,49,50,51,52,53,54]	-
↔ Body composition [7,10,35,36,41,46,47,48,49,50,51,52,53,54]	-
↓ Body fat [45,46,47,48,49,50,51,52,53,54]	-
↔ Bone formation [40,46,47,48,49,50,51,52,53,54]	-
↓ Bone resorption [40]	-
↑ LBM [10,35,36]	-
Attenuates the muscle mass decline [46,47,48,49,50,51,52,53,54]	-
↔ Circumference [10,35,36,46,47,48,49,50,51,52,53,54,56]	-
↔ Upper Limbs Volume [46,47,48,49,50,51,52,53,54]	-
↑ Balance [37]	-
↑ QoL [7,10,35,36,39,41,46,47,48,49,50,51,52,53,54,56]	-
↔ Depression [10,35,36,38]	-
↓ Depression [38]	-
↑ Self-perceptions [38,46,47,48,49,50,51,52,53,54]	-
↔ Norman score [46,47,48,49,50,51,52,53,54]	-
↓ Deterioration of physical function [46,47,48,49,50,51,52,53,54]	-
↔ DASH, BPI, FACT-B+4 or QLQ-BR23 [56]	-
↓ IGF-II levels [10,35,36]	-
↑ IGF-1 [45]	-
↓ TNF-α on their NK cells [7,41]	-
↔ miRNA [44]	-
Positive correlations between strength improvements and changes to circulating miRNAs [44]	-
↔ Telomere length [43]	-

BMI = body mass index; BMD = bone mineral density; DXA = dual x-ray absorptiometry; QoL = quality of life; LBM = lean body mass; ROM = range of motion; EMG = electromyographic; DASH = the disability of the arm, shoulder, and hand questionnaire; BPI = brief pain inventory; (FACT-B+4) = functional evaluation of chronic illness therapy breast cancer questionnaire; QLQ-BR23 = quality of life questionnaire module for breast cancer patients; SF-36 = short form questionnaire; VAS = visual analogue scales; PSPP = physical self-perception profile; BIRS = body image and relationships scale; DFS = disease free survival; OS = overall survival; DDFS = distant DFS; RFI = recurrence-free interval; IL-6 = interleukin-6; IL-1Ra = interleukin 1 receptor antagonist; IGF-II = insulin-like growth factor II; IGF-I = insulin-like growth factor I; TNF-α = tumor necrosis factor alpha; NK = natural killer; NKT = natural killer invariant; ↑ significant increase; ↓ significant decrease; ↔ without changes; (↑) no significant increase; NP = not reported.

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
