# Peer review of "Resistance Training in Breast Cancer Survivors: A Systematic Review of Exercise Programs"

_ijerph, 2020, doi:10.3390/ijerph17186511_

Round 1

Reviewer 1 Report

Thank you for the opportunity to read this well written, clear and concise study. It was interesting and a pleasure to read.

The introduction is very complete and allows the reader to know the main topic of the research, informs about the purpose and importance of the work in the clinical field, as well as responding to the question posed in the scientific context. It includes previous works on the subject in question and makes clear the aspects to be detailed in the review, which constitutes the object of the proposed research. It explains the general problem of the research, includes previous work on the subject in question and specifies the objective of the study.

The “methods” section is one of the most fundamental sections of a scientific article with these characteristics and it is well developed and organized. The research method used to locate the relevant studies is exhaustive, the design used for the selected articles is appropriate, a flow chart has been included that explains how many studies were initially identified, how many were selected, how many were excluded, and a validity of data has been carried out. the included studies. However, some aspects should be reviewed, as they could alter the veracity of the study, as follows:
- It would be convenient to add from and to the date on which the systematic search was carried out. This is important so that the reader knows the latest work on the subject in question.
- During the selection of articles, were there no duplicate articles? Please specify.
- It is recommended that an assessment of the risk of bias of the studies be carried out.
- Why was a meta-analysis not carried out? This factor would provide a summary effect measure of the included studies.

The results obtained are accurate, so the review can be characterized as having external validity and clinical importance. Although tables are presented more specifically to explain the different selected articles, it would be advisable to include a table that encompasses the main characteristics of the studies that make up the review: study, year and location; studied population, groups and study design; outcomes and measuring tools; intervention; measure time points, dropout and adverse effects; and main findings.

The conclusion section does not appear in the manuscript. Please add. It is essential in any scientific article to add the conclusion obtained from the different findings.

References

This section must be homogeneous. Please check it out.

Author Response

We appreciate the evaluation of the manuscript and the corrections you've pointed out to us and your constructive comments. Thank you very much.

We have corrected the errors you pointed out to us, and followed your recommendations. In relation to the conclusions, in the initial word document that was uploaded to the platform they were included but not in the pdf document, sorry. We have sent the English edition service of the publisher to correct the syntactic errors. Below we comment on the changes we have made in response to your kind suggestions.

At the suggestion of the English proofreaders, the title was: “Resistance training in breast cancer survivors: a systematic review: how it has been done”, has been modified to: “Resistance training in breast cancer survivors: a systematic review of exercise programs”

Comments indicated by the reviewer to respond or attend:

Point 1: It would be convenient to add from and to the date on which the systematic search was carried out. This is important so that the reader knows the latest work on the subject in question.

Response 1: We have added the search start and end dates of the articles to improve readers' interpretation of the work. These dates were from January 1, 1990 to December 6, 2019, (on pages 20, 86, 92, 248 and table 2)

Point 2:  During the selection of articles, were there no duplicate articles? Please specify.

Response 2: Although it was implicit in the “Flow chart” of searching and obtaining articles (figure 2), for better clarity we have pointed it out.

Point 3: It is recommended that an assessment of the risk of bias of the studies be carried out.

Response 3: According to its indication and to improve the work, we have made a review of the risk of bias, following the Cochrane tool RoB2, and we have added a figure with the graphic result, (figure 2). And throughout the text it has also been introduced (pages 21 and 285) and in the Item "2.3 Data Extraction..." (pages 259-265).

Point 4: Why was a meta-analysis not carried out? This factor would provide a summary effect measure of the included studies.

Response 4: We did not make meta-analysis, because the aim of the work was to analyze the characteristics of the training programs in clinical studies carried out in breast cancer supervising patients (frequency, load, series, muscle groups, training means, time of program implementation within the clinical phases, type of force (maximum force, power...) etc...). We do consider that the meta-analysis would be more convenient if the objective were to see the effect of the training (control group vs. experimental group and magnitude of change). But that was not our objective, and there are more meta-analyses in this respect. For us it was only a secondary objective to see which response variable had been monitored and very superficially to summarize the results observed in those monitored variables.

Point 5: The results obtained are accurate, so the review can be characterized as having external validity and clinical importance. Although tables are presented more specifically to explain the different selected articles, it would be advisable to include a table that encompasses the main characteristics of the studies that make up the review: study, year and location; studied population, groups and study design; outcomes and measuring tools; intervention; measure time points, dropout and adverse effects; and main findings.

Response 6: According to your indication, we have grouped the tables presented in the text, and these more grouped tables have been added as "Supplementary Materials available online”: Table 1: Characteristics of controlled trials reviewed; Table 2: Outcomes, measuring tools, and main findings, and Table 3: Exercise prescription.

Point 6: The conclusion section does not appear in the manuscript. Please add. It is essential in any scientific article to add the conclusion obtained from the different findings.

Response 6: Certainly it was a very big mistake in one of the files we sent on the first occasion, they are visible in all the documents we resent on this second occasion.

Point 7: References. This section must be homogeneous. Please check it out.

Response 7: we have reviewed this section, and revised it according to the style script that Mendeley has available for the “International Journal of Environmental Research and Public Health” format

Once again, thank you very much for your comments and observations which we believe have improved the quality of the work and the document.

Reviewer 2 Report

Introduction - No comments. Very well written and organized

Line 60-Spacing issues between "strength" and "that"

Page numbers are not in correct order as they repeat themselves.

Table 8 = UP not defined unless meant to be UB

More appropriate term for 15-20RM and 6-10RM may be high rep or low rep

Table 10. Volume of what? Not defined or discussed within text.

Lines 220 and 229 - Spacing between "strength" and "of"

Line 251 - "bodybuilding machines" is not an appropriate term. More appropriate would be strength-training machines

Line 280 - Resistance training is spelled incorrectly

Table 10 - Definitions - interleukin-6 is spelled incorrectly

Line 347 and 352 - space after strength before comma should be removed

There is not a conclusion written.

Author Response

We appreciate the evaluation of the manuscript and the corrections you've pointed out to us and your constructive comments. Thank you very much.

We have corrected the errors you pointed out to us, and followed your recommendations. In relation to the conclusions, in the initial word document that was uploaded to the platform they were included but not in the pdf document, sorry. We have sent the English edition service of the publisher to correct the syntactic errors. Below we comment on the changes we have made in response to your kind suggestions.

At the suggestion of the English proofreaders, the title was: “Resistance training in breast cancer survivors: a systematic review: how it has been done”, has been modified to: “Resistance training in breast cancer survivors: a systematic review of exercise programs”

Comments indicated by the reviewer to respond or attend:

Point 1: Line 60-Spacing issues between "strength" and "that"

Response 1: The error has been corrected.

Point 2: Page numbers are not in correct order as they repeat themselves.

Response 2: Pagination has been corrected and modified (it is formatted for printing, but can be changed to inline format : white or not between portrait to landscape and landscape to portrait)

Point 3: Table 8 = UP not defined unless meant to be UB

Response 3: The error has been corrected by removing UP and placing UB

Point 4: More appropriate term for 15-20RM and 6-10RM may be high rep or low rep

Response 4: It has been modified, indicating only the repetitions without qualifying them qualitatively.

Point 5: Table 10. Volume of what? Not defined or discussed within text.

Response 5: It was not clear, we have modified it by writing: "Upper Limbs Volume"

Point 6: Lines 220 and 229 - Spacing between "strength" and "of"

Response 6: The error has been corrected.

Point 7: Line 251 - "bodybuilding machines" is not an appropriate term. More appropriate

would be strength-training machines

Response 7: Has been replaced throughout the text by : "strength-training machines"

Point 8: Line 280 - Resistance training is spelled incorrectly

Response 8: The error has been corrected.

Point 9: Table 10 - Definitions - interleukin-6 is spelled incorrectly

Response 9: The error has been corrected.

Point 10: Line 347 and 352 - space after strength before comma should be removed

Response 10:The error has been corrected.

Point 11: There is not a conclusion written.

Response 11: Certainly it was a very big mistake in one of the files we sent on the first occasion, they are visible in all the documents we resent on this second occasion.

Once again, thank you very much for your comments and amendments.

Round 2

Reviewer 2 Report

Revisions have been made and addressed appropriately. No further revisions recommended.